# Artificial Intelligent Deep Learning Molecular Generative Modeling of Scaffold-Focused and Cannabinoid CB2 Target-Specific Small-Molecule Sublibraries

**DOI:** 10.3390/cells11050915

**Published:** 2022-03-07

**Authors:** Yuemin Bian, Xiang-Qun Xie

**Affiliations:** 1Department of Pharmaceutical Sciences and Computational Chemical Genomics Screening Center, Pharmacometrics & System Pharmacology PharmacoAnalytics, School of Pharmacy, University of Pittsburgh, Pittsburgh, PA 15261, USA; yuemin.bian@pitt.edu; 2NIH National Center of Excellence for Computational Drug Abuse Research (CDAR), University of Pittsburgh, Pittsburgh, PA 15261, USA; 3Drug Discovery Institute, University of Pittsburgh, Pittsburgh, PA 15261, USA; 4Departments of Computational Biology and Structural Biology, School of Medicine, University of Pittsburgh, Pittsburgh, PA 15261, USA

**Keywords:** drug discovery, generative modeling, recurrent neural network, deep-learning molecule generation model (DeepMGM), cannabinoid receptor 2 (CB2), negative allosteric modulator (NAM)

## Abstract

Design and generation of high-quality target- and scaffold-specific small molecules is an important strategy for the discovery of unique and potent bioactive drug molecules. To achieve this goal, authors have developed the deep-learning molecule generation model (DeepMGM) and applied it for the de novo molecular generation of scaffold-focused small-molecule libraries. In this study, a recurrent neural network (RNN) using long short-term memory (LSTM) units was trained with drug-like molecules to result in a general model (g-DeepMGM). Sampling practices on indole and purine scaffolds illustrate the feasibility of creating scaffold-focused chemical libraries based on machine intelligence. Subsequently, a target-specific model (t-DeepMGM) for cannabinoid receptor 2 (CB2) was constructed following the transfer learning process of known CB2 ligands. Sampling outcomes can present similar properties to the reported active molecules. Finally, a discriminator was trained and attached to the DeepMGM to result in an in silico molecular design-test circle. Medicinal chemistry synthesis and biological validation was performed to further investigate the generation outcome, showing that XIE9137 was identified as a potential allosteric modulator of CB2. This study demonstrates how recent progress in deep learning intelligence can benefit drug discovery, especially in de novo molecular design and chemical library generation.

## 1. Introduction

Drug discovery is an expensive and lengthy process. It can take over 12 years to develop a new drug, with an average cost of USD 2.6 billion [1]. By screening compounds in large volumes, high-throughput screening (HTS) techniques provide a solution for lead identification of bioactive molecules [2]. However, the estimated number of synthetic feasible compounds with drug-like properties now reaches 10^60^, which represents an enormous chemical space that can hardly be exhausted [3,4]. Even with the recent progress in HTS techniques, finding a bioactive molecule for a given target is similar to finding a needle in a haystack, if not harder. The concept of privileged scaffolds was first proposed by Evans et al., in 1988 [5]. Privileged scaffolds present in molecules with affinities to different receptors [6,7]. Generating compound collections with a focus on privileged scaffolds turns out to be a solution for the discovery of unique and potent bioactive molecules [8]. The following question arises naturally: is there an efficient and effective approach to building a scaffold-focused chemical library? We would like to present a putative answer in the artificial intelligence era with this manuscript: deep learning generative modeling.

Cheminformatic drug discovery is an area of research enriched in research [9,10]. The capability of dealing with large datasets to detect hidden patterns and facilitate future data prediction in a time-efficient manner favored the emergence of machine learning [11,12,13]. In recent years, the number of expeditions toward generative chemistry has mushroomed [14,15,16,17,18,19,20,21,22,23] and they have explored the possibility of utilizing generative models to design molecular structures with desired properties. From the training molecules, generative models study and summarize a probability distribution to sample new molecules that are similar to the training data [24,25]. More details about recent advances were previously reviewed [11]. To apply generative modeling for building scaffold-focused chemical libraries, three objectives were investigated in this study.

Objective one: could general drug-like molecules be generated from starting scaffolds? A general molecule generation model (g-DeepMGM) was designed and constructed by adapting deep-learning recurrent neural networks (RNN) with long short-term memory (LSTM) [26] units. The g-DeepMGM was iteratively trained with a simplified molecular-input line-entry system (SMILES) strings of collected training molecules to learn: (1) the grammar of composing a valid string and (2) the properties of a drug-like molecule. Two sampling practices using indole and purine scaffolds for compounds generation were assessed and discussed. 

Objective two: could scaffold-focused chemical libraries be target-focused as well? A target-specific molecule generation model (t-DeepMGM) was developed after the transfer learning [26,27] of the pre-trained g-DeepMGM using bioactive molecules for a given target, cannabinoid receptor 2 (CB2) for instance. The transfer learning process fine-tuned the model to combine the general features learned to handle a specific data structure. 

Objective three: could this study result in a close-loop design-test circle? A multilayer perceptron-based discriminator was trained to distinguish active CB2 ligands from inactive and random molecules. The sampling outcome from the t-DeepMGM was evaluated and scored by this discriminator. The combination of the DeepMGM and the discriminator can potentially contribute to an in silico molecule design-test loop.

As the target protein for the t-DeepMGM, cannabinoid receptors, which have two main subtypes named CB1 and CB2, belong to the class A rhodopsin-like GPCR family [28,29]. Unlike CB1, which is enriched in the central nervous system (CNS) [30], CB2 is mostly distributed in the peripheral parts, including the immune system and hematopoietic cells [31]. Studies have disclosed that CB2 may be expressed in the CNS as well [32]. Selective regulation of CB2 exhibits therapeutic potential for inflammatory pain, autoimmune disorders, as well as osteoporosis [33]. Classical GPCR drug discovery focuses on developing molecules towards orthosteric binding pockets, which results in corresponding protein-specific agonists, antagonists, inverse agonists, etc. Recent advancements in GPCR conformational studies have largely facilitated the identification of distinct binding sites (allosteric binding pockets) [34]. Developing ligands that target allosteric sites to modulate pharmacological functions brings about new opportunities [35].

Altogether, this study demonstrates an innovative framework for using deep learning to facilitate de novo molecular design as well as the generation of focused chemical libraries. Generative modeling is a positive addition to modern medicinal and computational chemistry approaches [36,37]. We are optimistic to expect more fruitful applications to be reported in the near future.

## 2. Materials and Methods

### 2.1. Dataset Preparation

A total 500,000 molecules were randomly collected from the ZINC database [38] for training the general molecule generation model. With the goal of covering a broad chemical space, both drug-like and non-drug-like compounds with extensively diversified scaffolds were included in the collection. From the perspective of Lipinski’s rule of five (RO5) [39], 92.9% of the collected compounds have a molecular weight less than 500, 91.9% have a LogP less than 5, 99.0% have less than 10 hydrogen bond acceptors, and 99.4% have less than 5 hydrogen bond donors. A total of 87.2% of the collected compounds fulfill all criteria of the RO5. There are two major reasons to select compounds from the ZINC database for collection in this study. First, ZINC is constituted with commercially available compounds, which indicates synthetic feasibility to some extent. Second, compounds in this database have not necessarily reported biological activities, which enable the inclusion of novel structures to further expand the chemical space. 

A total of 949 molecules with reported *K_i_* values for cannabinoid receptor 2 were collected from the ChEMBL database [40] for the transfer learning of a target-specific molecule generation model. A total of 385 compounds have reported *K_i_* values less than 100 nM, while 564 compounds have higher values. Combining moderate and weak binders helps increase the structural diversity by introducing various moieties. As a molecule generation campaign for a given target, compounds with diversified structural features are preferred.

### 2.2. Molecular Representation and One-Hot Encoding

SMILES strings are used as the molecular representation to describe the compounds. SMILES, which is the abbreviation for simplified molecular-input line-entry system [41,42], is a specification in the form of a line notation for describing the structure of chemical species using short ASCII strings. One molecule can be written into multiple SMILES strings with different starting atoms and directions. This feature is utilized in this study to describe a lead scaffold fragment in multiple SMILES strings that generated moieties that can be linked to individual bondable atoms. To transfer the SMILES strings into a machine-readable format, one-hot encoding [43] was adapted to present molecules as one-hot vectors. With one-hot encoding, a binary vector is established with the size of the number of unique characters in SMILES strings. For each character in one SMILES string, a specified integer index *i* is assigned. The binary vector for this character is constructed with all zeros but one for the *i*th entry. For a simplified example, three unique characters can be used: “C”, “O”, and “N”. Input “C” is transferred to (1, 0, 0), “O” to (0, 1, 0), and “N” to (0, 0, 1) after one-hot encoding. In the training process of this study, each SMILES string is started with the character “G” and finished with character “E” to denote a complete string. In the process of sampling, the ending character “E” also indicates a complete generation of one molecule. There are a total of 29 unique characters in this study that have been one-hot encoded.

### 2.3. Neural Network Implementation

Recurrent neural networks with LSTM units for both the g-DeepMGM and t-DeepMGM were built, trained, and fine-tuned using the Python module Keras [44] with TensorFlow [45] as the backend. Models are built to be sequential so as to have a linear stack of layers. The function *LSTM* was used to create a long short-term memory layer. The hyperparameters including units (*256* and *512*), activation (*tanh*), and recurrent_activation (*sigmoid*) were tested. The function *Dropout* was applied to add a dropout layer. The hyperparameter dropout rate (*0.2, 0.3, 0.4, 0.5*) was optimized. The function *Dense* was used to create a densely-connected layer. The hyperparameters, including units and activation (*relu*, *softmax*, *tanh*), were tuned. The g-DeepMGM was comprised of four layers totaling 825,629 trainable parameters: layer 1, an LSTM with 256 units; layer 2, a dropout with the dropout rate of 0.3; layer 3, an LSTM with 256 units; layer 4, a fully connected layer with an activation function of *softmax*. The *categorical cross-entropy* was selected as the loss function and the *Adam* method was selected as the optimizer for training. *Adam* optimization is a stochastic gradient descent method that is based on adaptive estimation of first-order and second-order moments. Log-likelihood was calculated using the function of *log_loss* in the python module sklearn [46]. Wasserstein distance was calculated using the function of *wasserstein_distance* in the python module scipy. The t-DeepMGM was comprised of the same architecture as the g-DeepMGM. The transfer learning process fine-tuned a pre-trained g-DeepMGM with a small set of target-specific molecules to result in a functional t-DeepMGM. 

### 2.4. Transfer Learning

The purpose of transfer learning is to refine the molecules generated by the g-DeepMGM to have the potential for target specificity. In this study, the g-DeepMGM is trained with a large collection of a half million molecules from the ZINC database to grasp the grammar of composing a valid SMILES string for a drug-like molecule. Then, the model is refined with 949 CB2 ligands to result in the t-DeepMGM. The process of transfer learning empowers the model to combine the general features learned from the large dataset to handle a specific data structure of a smaller dataset. During the transfer learning process, the first LSTM layer of the g-DeepMGM was frozen to keep the parameters constant. The second LSTM layer and the dense layer allowed their parameters to be changed to adjust the confronting input data. Appendix A shows the transfer learning strategies with different dataset sizes and similarities.

### 2.5. Model Evaluation

The sampling results were investigated through three aspects: validity, uniqueness, and novelty. Validity evaluates the percentage of generated SMILES strings that can be conveyed back to molecular structures. Among valid SMILES strings, uniqueness assesses the percentage of strings that are mutually different. Among valid and unique SMILES strings, novelty checks the percentage of strings that are not simply recurring in the training set. The similarity scores calculated based on Molecular ACCess System (MACCS) fingerprints [47] were used to assess the novelty. Sampled molecules with a similarity score of less than 0.85 were considered novel molecules. The sampling temperature affects the conditional probability distribution of the characters in the SMILES vocabulary. Typically, elevated validity, but flat uniqueness and novelty, can be expected under a lower sampling temperature. In contrast, a higher sampling temperature can prompt flat validity, but elevated uniqueness and novelty. 

The synthetic accessibility (SA) score was calculated to evaluate the synthetic feasibility of generated molecules. The RDKit [48] contribution, *SA_Score*, was used for calculating the SA scores. The quantitative estimate of druglikeness (QED) was computed to evaluate the druglikeness of generated molecules. The function QED from the Python package *silicos_it* [49] was used to perform the calculation. 

A t-distributed stochastic neighbor embedding (t-SNE) [50] algorithm was used as a dimensionality reduction method to visualize high-dimensional physical–chemical properties and molecular fingerprints in a two-dimensional chart. The t-SNE plot provides intuitional ideas about the coverage of the chemical space and shows whether the scaffold-focused and the target-focused library generation can have concentrated property distribution. There are two stages in the t-SNE algorithm: (1) the probability distribution over pairs of high-dimensional features is constructed, and (2), a similar probability distribution over the points in the low-dimensional map is defined. The process minimizes the Kullback–Leibler divergence (KL divergence) between the two distributions. The Python library scikit-learn [46] was applied for t-SNE analysis and matplotlib [51] was used for plotting.

The distribution and the correlation of four representative physical–chemical properties were plotted for both the training and sampled molecules for comparison. The four representative physical–chemical properties are molecular weight (M.W.), topological polar surface area (TPSA), molecular refractivity (SMR), and octanol-water partition coefficient (SlogP). Physical–chemical properties and MACCS fingerprints were calculated using the Python library RDKit. MACCS fingerprints include 166 substructure keys, each of which represents the existence of a certain substructure feature.

### 2.6. MLP Discriminator for Cannabinoid Receptor 2

A multilayer perceptron (MLP) based discriminator is attached in conjunction with the t-DeepMGM after the generation of a compound library for a given protein. The purpose of this is to predict the likelihood of generated compounds to be active toward the biological target. The discriminator is composed to have three densely connected hidden layers with 166 hidden units for each layer. The solver function *Adam* is set to optimize weights. The activation function is set to *relu*. CB2 compounds with reported *K_i_* values were collected from the ChEMBL database. Using the cutoff of 100 nM, active and inactive molecules were separated. Random drug-like molecules that function as decoys were mixed with inactive molecules afterwards. The discriminator was trained to classify and distinguish active molecules from inactive and random chemicals. The Python library scikit-learn was used for model training, data prediction, and interpretation of results. The development process is in accordance with our previous study on building supervised machine learning classifiers [12].

### 2.7. Chemistry

All reagents were purchased from commercial sources and used without further purification. Analytical thin-layer chromatography (TLC) was performed on SiO_2_ plates on glass. Visualization was accomplished by UV irradiation at 254 nm. Flash column chromatography was performed using the Biotage Isolera flash purification system (Biotage, Uppsala, Sweden) with SiO_2_ 60 (particle size 0.040−0.055 mm, 230−400 mesh). NMR was recorded on a Bruker 600 MHz spectrometer (Bruker, Billerica, MA, US). Splitting patterns are indicated as follows: s—singlet; d—doublet; t—triplet; m—multiplet; br—broad peak. The starting material, 1-methyl-1H-benzo[c][1,2]thiazin-4(3H)-one 2,2-dioxide, was purchased from the Enamine database (Enamine, Monmouth Jct., NJ, US) and was used as received. A 1 mL V-Vial (Fisher Scientific, Waltham, MA, US) with an open-top screw cap was used to perform reactions. The ethanol was reagent alcohol (ethanol (88 to 91%), methanol (4.0 to 5.0% *v/v*), isopropyl alcohol (4.5 to 5.5%)) in a 4 L poly bottle purchased from Fisher Chemical (Fisher Scientific, Waltham, MA, US). The magnetic stirring bar was octagonal with a diameter of 2 mm and a length of 7 mm. A Fisherbrand porcelain Büchner funnel (Fisher Scientific, Waltham, MA, US) (with plate diameter of 43 mm) was used for suction filtration applications. Fisherbrand class A clear glass threaded vials with caps (24 mL, catalog number: 14955318) were used to save target compounds. 

The production of (*E*)-3-((benzylamino)methylene)-1-methyl-1H-benzo[c][1,2]thiazin-4(3H)-one 2,2- dioxide (1–37) is as follows. A 1.5 mL vial equipped with a stir bar is charged with 1-methyl-1H- benzo[c][1,2]thiazin-4(3H)-one 2,2-dioxide (105 mg, 0.498 mmol). Triethyl orthoformate (0.10 mL, 0.615 mmol) and benzylamine (0.05 mL, 0.474 mmol) were added via syringe sequentially. The vial was capped and placed in an oil bath. The temperature of the oil bath was rapidly increased to 130 °C before adding ethanol (0.2 mL) via a syringe. After 2 h, the vial was removed from the oil bath and the solution was allowed to cool to the room temperature (rt). The reaction mixture was concentrated by rotary evaporation and purified by silica gel flash column chromatography eluting with 0–35% ethyl acetate in hexanes to yield 1–37 (58 mg, 37%). ^1^H NMR (600 MHz, CDCl_3_) δ 10.88 (s, 1H), 8.08–8.10 (m, 1H), 7.92 (d, J = 13.5 Hz, 1H), 7.56–7.59 (m, 1H), 7.39–7.42 (m, 2H), 7.36–7.37 (m, 1H), 7.28–7.30 (m, 2H), 7.23–7.25 (m, 1H), 7.15 (d, J = 8.2 Hz, 1H), 4.61 (d, J = 5.7 Hz, 2H), 3.35 (s, 3H). ^13^C NMR (150 MHz, CDCl_3_) δ 181.07, 153.31, 153.10, 142.75, 135.22, 134.43, 129.39, 128.83, 128.61, 127.76, 125.40, 124.01, 118.37, 108.48, 54.12, 53.96, 32.37. NMR spectra are available as Appendix A.

### 2.8. CB2 [^35^S]-GTPγS Functional Assay

SPA [^35^S]-GTPγS experiments were conducted by EuroscreenFast (Charleroi, Belgium) with Epics Therapeutics’ (Charleroi, Belgium) membrane preparations. Human CB2 receptor (FAST-041G) membrane extracts were prepared from CHO-K1 cells overexpressing recombinant human CB2 receptor, diluted in assay buffer (20 mM HEPES pH 7.4, 100–200 mM NaCl, 10 μg/mL saponin, MgCl_2_ at optimized concentration for the specified receptor, 0–0.1% BSA), and kept on ice. The membranes were mixed with GDP and incubated for at least 15 min on ice. In parallel, GTPγ[^35^S] was mixed with the beads just before starting the reaction. The following reagents were successively added in the wells of an Optiplate (Perkin Elmer, Waltham, MA, USA): 50 μL of test or reference ligand; 20 μL of the membranes; GDP mix; after a 10 min incubation at RT, 10 μL of reference agonist (CP55,940) at historical EC_20_ (to initially test for CB2 PAM activity); and 20 μL of the GTPγ[^35^S] beads mix. The plates were covered with a top seal, mixed on an orbital shaker for 2 min, and then incubated for 1 h at room temperature. Then, the plates were centrifuged for 10 min at 2000 rpm, incubated at room temperature for 1 h, and counted at 1 min/well with a Perkin Elmer TopCount reader.

Dose-response data from test compounds were analyzed with XLfit (IDBS) software using a nonlinear regression applied to a sigmoidal dose-response model. Agonist activity of reference CP55,940 is expressed as a percentage of the activity of the reference agonist at its EC_100_ concentration. NAM activity of test compounds is expressed as a percentage of the activity of the reference agonist at its EC_80_ (0% inhibition) concentrations.

## 3. Results and Discussion

### 3.1. The Overview of General/Target-Specific Molecule Generation Models (g-DeepMGM and t-DeepMGM)

Figure 1 gives a schematic illustration of the pipeline for model training and sampling in this study. A total of 500,000 molecules with diversified scaffolds were randomly collected from the ZINC database for g-DeepMGM model training (Figure 1a). In total, 87.2% of collected molecules follow the RO5, while the remaining 12.8% have one or more violations of these drug-likeness rules. SMILES strings were used as the molecular representation. SMILES describes molecules with an alphabet of characters following a peculiar grammar. In order to sample valid SMILES strings as the outcome, the generative model should learn and grasp the grammar of using SMILES to compose a molecule. The large-scale training collection of compounds provides learning resources that enable the generative model to comprehend after iterative learning. Meanwhile, the inclusion of both drug-like and non-drug-like structures further expands the covered chemical space. The expanded chemical space then accredits the model to have exposure to different possibilities of combining moieties with distinctive properties. To transfer a SMILES string to a machine-readable format, a binary vector can be constructed with one-hot encoding. The ending character “E” was added to each SMILES string to denote the completion of the structure. With the one-hot encoded chemical data as the input, the g-DeepMGM, an RNN with LSTM layers, was trained to predict the probability distribution of the *n*+1th character given the input of string with *n* characters. The training process was monitored using categorical cross-entropy as the loss function. With the well-trained g-DeepMGM, the molecule generation process can start with a given starting scaffold to sample the next character until the ending character “E” is reached. 

The t-DeepMGM for CB2 was built after the transfer learning of the established g-DeepMGM with collected bioactive CB2 ligands (Figure 1b). A total of 949 molecules with reported CB2 *K_i_* values were collected from the ChEMBL database and one-hot encoded into binary vectors. Compared to the half million compounds involved in g-DeepMGM training, the CB2collection of compounds is smaller in size and different in chemical properties, with focused scaffolds and limited coverage of chemical space. Transfer learning empowers the model to combine the general features learned from the large dataset to handle a specific data structure of a smaller dataset. With the established t-DeepMGM, five seed scaffolds were given for molecular sampling. The sampled outcome is supposed to be specifically associated with the CB2 target. An MLP-based discriminator was constructed to investigate compounds sampled by the t-DeepMGM (Figure 1c). The MACCS fingerprints were calculated as the molecular representation. The discriminator was trained to distinguish CB2 active compounds from inactive and random molecules.

### 3.2. RNN Model Training and Sampling

The g-DeepMGM model is constructed with two LSTM layers and one dense layer for outputting SMILES characters (Figure 2a). The half million ZINC compounds were divided into a training set and a validation set following an 80% to 20% ratio. Eventually, 100 training epochs using mini-batches were conducted with a batch size of 512 molecules. The trained model at each epoch was saved, and the loss and the accuracy calculated. Figure 2b,c illustrates the plotted loss and accuracy values at each training epoch. Exhibiting the performance of models with 256 units (Appendix A) and 512 units (Appendix A) for each LSTM layer demonstrates one example of hyper-parameter fine-tuning. The convergence of both the training loss and the validation loss can be observed for the model with 256 units (Figure 2b). The loss tends to be steady after a training process of around 40 epochs. The same trend is observed in the accuracy plot, in which the constant accuracy score is achieved after training for around 40 epochs (Figure 2c). For the model with 512 units, clear over-fitting is indicated by the perceptible gap between the training loss and the validation loss. After the first 10 epochs, the increase of validation loss is spotted along with the decrease of training loss. The 512 unit model is an architecture with more adjustable parameters than can be justified by the input data. On the accuracy plot, despite the climbing training accuracy, the validation accuracy turns out to stay at the same level as the model with 256 units per LSTM layer. The log-likelihood and Wasserstein distance (Appendix A) are two additional metrics for evaluating model performance at different epochs. The observation is consistent that the model with 512 units suffers from a more severe over-fitting issue, while steady values are spotted for the model with 256 units after epoch 40. 

The training process results in a probability distribution of the next SMILES character given the input string. The sampling process then predicts the next character in sequence, sticking to the resulted probability distribution. Then, that character is added to the input for the next step of the prediction. The process is iterated until the ending character is reached. To generate a scaffold-focused compound library, the SMILES string of the lead scaffold is supplied as the input to initiate the sampling of the well-trained g-DeepMGM model. Using the indole scaffold as one example (Figure 2a), to add moieties to the carbon at position 3 the SMILES string “C12=C(C=CC=C2)NC=C1” is fed to the g-DeepMGM in step 1 with the character “C” as the output. The character “C” is then added to the initial string to function as the input in step 2 with the character “N” as the output. Again, “N” is added to function as the input in the next step. This character sampling process is continued until the ending character “E” is reached, which suggests the completed sampling of a molecule. There are 7 possible modification positions on an indole ring. By selecting different starting and ending atoms to compose SMILES strings, moieties can be added precisely [52]. In this study, sampling temperatures have been investigated as well. A lower temperature makes the model prefer a more conservative prediction for the next character, while a higher temperature prefers diversified character selection. The sampling temperature affects the conditional probability distribution of the characters in the SMILES vocabulary.

### 3.3. Indole Scaffold Compounds Generation

Building chemical libraries with privileged scaffolds is an emerging strategy for identifying biologically active molecules [8,53]. Drug discovery stories have revealed that high-affinity ligands toward various targets can be constructed from certain scaffolds (Figure 3a). Generating compounds with a focus on privileged scaffolds turns out to be a solution for discovering unique and potent bioactive molecules. In this section, the indole is selected as an example of a privileged scaffold. The capability of the g-DeepMGM to generate an indole-focused compound library was investigated. 

The g-DeepMGM at four training epochs, 10, 20, 40, and 100, were selected for independent molecular sampling as an evaluation of the training progress. Four sampling temperatures, 0.5, 1.0, 1.2, and 1.5, were applied to inspect the influence of the altered probability distribution on sampling results. The indole scaffold has seven possible positions for adding moieties. Seven SMILES strings reflecting these addition positions were prepared and used as the initial input to feed to the g-DeepMGM. For each training epoch amount (total four) at each sampling temperature (total four), the g-DeepMGM sampled 2000 output SMILES strings for every addition position (total seven) on the indole. Thus, 224,000 SMILES strings were sampled. The sampled results were examined from three aspects: (1) whether the generated SMILES strings are valid, so that they can be converted to structures (validity), (2) whether valid SMILES strings are unique from each other (uniqueness), and (3) whether valid and unique SMILES strings are novel and different from the half million molecules in the training set (novelty).

The differences among the validity, uniqueness, and novelty scores for molecules generated at different epochs and temperatures can be observed (Table 1). From epoch 10 to 100, the ability of the g-DeepMGM to generate valid SMILES strings increased and peaked at epoch 40. The training process enables the model to teach itself the proper grammar of composing the SMILES of a general drug-like molecule. It is supposed that the model should illustrate increased sophistication with an increased number of training epochs. At epoch 10, the g-DeepMGM can already have 86.7% of the sampled strings valid at temperature 0.5, as the first 10 epochs underwent the most drastic improvements to the loss and accuracy (Figure 2b,c). The validity drops at epoch 100, regardless of showing the lowest training loss. This observation is indicative of the over-fitting of the g-DeepMGM. It is not necessary to scan a half million compounds with mini-batches 100 times. The loss and the accuracy for the validation set maintained steady since epoch 40. From temperatures 0.5 to 1.5, a clear trend of decreasing validity and increasing both uniqueness and novelty can be spotted. Taking the g-DeepMGM at epoch 40 as one example, the validity decreased from 90.5% at temperature 0.5 to 39.9% at temperature 1.5, while the uniqueness and novelty increased from 8.6% and 20.2% to 47.6% and 65.3% respectively. At a lower temperature, conservative predictions that are closely associated with the probability distribution of SMILES characters are more likely to result in valid but repeated strings. In contrast, at a higher temperature, the prediction for the next SMILES character is more aggressive, which can result in unique but sometimes invalid strings.

Figure 3b demonstrates an example of the generation outcomes of the g-DeepMGM at epoch 40 under each sampling temperature, for seven addition positions of the indole. At temperature 0.5, instead of only generating simple aliphatic chains, ring systems including cyclohexane rings, phenyl rings, and piperazine rings can also be sampled. Primary, secondary, and tertiary amines, as well as methoxy groups, exist in the structures. Limited types of atoms can also be noticed: only carbon, nitrogen, and oxygen atoms are involved. At temperature 1.0, improved atom diversity is an obvious change. The addition of fluorine, bromine, and even a sulfonyl group substantially enlarged the chemical space that the generated molecules can cover. It is not uncommon to note that simple aliphatic carbon chains can output from the g-DeepMGM, even at a higher sampling temperature. At temperature 1.2, a heptyl chain can still be generated. More interestingly, a positively charged quaternary ammonium cation can be observed. Quaternary ammonium cations are permanently charged, independent of the pH of their solution. The generation of charged molecules in addition to the neutral compounds may further expand the application scope of the g-DeepMGM. At temperature 1.5, sampled molecules are composed of diversified atoms and various functional groups including pyrimidine, quinuclidine, naphthalene, etc. For observation of the structural variety, a similarity search on SciFinder was conducted, and reported compounds with similar structures have been identified. The creative outcomes of the g-DeepMGM at temperature 1.5 still fall in the existing chemistry knowledge framework. The synthetic feasibility for generated molecules at different temperatures was evaluated using the synthetic accessibility (SA) score. The SA score is calculated based on fragment contributions and a complex penalty where historical synthetic knowledge and non-standard structural features were captured [54]. Generated molecules have their SA scores mostly enriched between 1.5 and 3.5 (Appendix A), which is indicative of good synthetic feasibility. The quantitative estimate of druglikeness (QED) was computed. QED functions are based on the underlying distribution data of drug properties [55]. The distribution of QED scores for the generated molecules is clearly left-skewed, with modes around 0.7, indicating satisfactory druglikeness properties (Appendix A). 

Generated compounds from the seven addition positions at epoch 40 at all four sampling temperatures were combined. To evaluate how different the generated indole molecules were from the half million training compounds, both physical–chemical properties and MACCS fingerprints were calculated for comparison. The t-SNE analysis was conducted to reduce the dimension of features to two, as the first two components could explain more than 80% of the variance (Appendix A). On the t-SNE plot for physical–chemical properties (Figure 3c and Appendix A), the blue dots are from training molecules that give the background of overall property space. The colorful dots are from generated indole molecules that spread mostly within the boundary defined by blue dots. The distribution of the colorful dots has focused areas, rather than generally overlapping with the blue dots. This is consistent with the purpose of generating a scaffold-focused (indole-focused in this case) chemical library. Using the indole as the scaffold for molecule generation introduces a pre-existing bias that certain chemical space will be emphasized, while others will be less prioritized. Similarly, the MACCS-based t-SNE analysis confirmed the finding (Figure 3d and Appendix A). 

Another privileged scaffold, purine, was selected for evaluation to further investigate the generation of the scaffold-focused chemical library with the g-DeepMGM. The result is detailed in the Appendix A.

Besides the t-SNE analysis, which embedded high-dimensional data into a 2-dimensional space for visualization, four specific molecular descriptors, M.W., TPSA, SMR, and SlogP were selected for direct comparison. The distribution of these descriptors and the correlation of each pair were plotted for the training molecules and the generated indole and purine molecules (Figure 4). Generally, the assessment of these descriptors further reflects that these three chemical collections are distinct from each other. Distributions of the M.W. for generated indole (mean = 260.88, SD = 63.99) and purine (mean = 271.74, SD = 64.75) compounds overlap well, and to the left of the training molecules (mean = 365.80, SD = 80.23). The generated molecules are smaller in size, which implies the possibility of adding additional moieties onto other addition positions on the scaffold. Distinctive distributions of the TPSA can be observed. The TPSA of the training molecules has the biggest mean (73.33) and the widest spread (SD = 80.23). Generated purine molecules are prone to have a confined TPSA range (mean = 68.78, SD = 16.51), while generated indole molecules with low TPSA values are enriched (mean = 38.57, SD = 20.30). For the SMR, the distribution from generated purine molecules (mean = 76.69, SD = 18.50) mostly overlaps with generated indole molecules (mean = 79.51, SD = 18.23), with a smaller mean value. The distribution from training molecules is to the right, with a bigger mean (98.79) and a standard deviation of 22.07. The distribution of the SlogP for the generated indole compounds shares a close mean (3.37) but a smaller SD (1.17) to the training molecules (mean = 3.09, SD = 1.41). The distribution of the SlogP for generated purine molecules (mean = 2.05, SD = 1.20) possesses a smaller mean to the left of indole and training compounds. The comparison using these four molecular descriptors exemplifies that generated scaffold-focused chemical libraries do not simply mimic the properties exhibited from the training molecules. With lower SD values for distributions of properties, generated scaffold-focused molecules can be concentrated to emphasize a certain chemical space.

### 3.4. Transfer Learning for Cannabinoid Receptor 2

Using the g-DeepMGM to generate scaffold-focused chemical libraries is a feasible solution for discovering unique and potent bioactive molecules. To push one step further from sampling general drug-like chemicals, generating scaffold-focused molecules with target specificity can be of interest in target-driven small molecule drug discovery. A target-specific molecule generation model (t-DeepMGM) was proposed and trained after the transfer learning process of the g-DeepMGM. The purpose of transfer learning is to refine the molecule generation from the g-DeepMGM to have potential target specificity. There are reported applications of using transfer learning to generate focused libraries [52,56,57]. Appendix A demonstrates transfer learning strategies in facing of different data structures of the transfer learning set. In this practice, 949 molecules with reported *K_i_* values were collected to form the transfer learning set. Comparing it to the half million ZINC compounds involved in the training of the g-DeepMGM, this transfer learning set is smaller in size and structurally different. The g-DeepMGM was fine-tuned on this transfer learning set with one or two LSTM layers locked for comparison. In addition to using the same batch size of 512 as the g-DeepMGM for training, a smaller batch size of 128 was evaluated as well (Figure 5a). A total of 40 training epochs were conducted. A lower loss and a higher accuracy were achieved by locking one LSTM layer (Figure 5b–e). Also, fine-tuning with a small mini-batch of 128 molecules outperformed a larger batch of 512 molecules. 

The t-DeepMGM that trained with a batch size of 128 and with one LSTM layer locked was applied for molecule generation at epoch 20 at temperature 1.2. Four well-known CB2 scaffolds were used as the input to initiate the sampling process. Figure 5f exhibits sampling examples in comparison with known CB2 active ligands. With the JTE-907 [58,59] scaffold, the amide side chain was cut for molecule generation. The amide bond can be reconstructed as shown by compound 1. The length of the side chain can vary from no carbon (compound 1) to three carbon atoms (compound 2). Alternatively, cyclohexane (compound 1), morpholine (compound 2), and pyrrolidine (compound 3) rings were generated as the benzodioxole group. JTE-907 has not recurred. 

The biamide CB2 inverse agonist was previously developed by our group [60]. The t-DeepMGM was applied to sample side chains for the aromatic ring with C in the middle. Structurally diversified moieties such as isopentane (compound 4), cyclopropylamine (compound 5), and methoxypropane (compound 6) were constructed. Notably, the size of the diethylamine group on the known CB2 ligand is similar to these generated moieties.

AM630 is a classic CB2 antagonist [61,62]. Modifications to this compound are mainly focused on positions 1 and 3 of the indole scaffold. The sampling practices on these two positions were performed. With the N-ethylmorpholine group attached to the position 1, the t-DeepMGM sampled pyrrolidine (compound 7), morpholine (compound 8), and ethyl bromide (compound 9) to the position 3. Both ring-based and chain-based moieties are observable. With the benzaldehyde moiety attached at the position 3, compounds 10 to 12 were sampled. In particular, the compound 10 has one more connecting carbon between the indole and the morpholine than the AM630. The compound 12 was composed of a thiomorpholine to replace the morpholine while both the sulfur and oxygen atoms were capable of forming hydrophilic interactions. 

The fourth scaffold comes from triaryl sulfonamide derivatives that were previously developed by our group as CB2 inverse agonists [63]. The t-DeepMGM was adapted to sample moieties from the nitrogen of the sulfonamide group. The sulfonamide group has recurred in the generated compound 13 with a pyridine ring connected. Unconventional moieties such as S-methyl-methanesulfinamide group (compound 15) can result from the sampling process as well.

Generated molecules for CB2 have their SA scores enriched between 2 and 3 (Appendix Aa), which indicates good synthetic feasibility. Also, generated molecules are quite acceptable from the perspective of druglikeness with the mean QED = 0.61 (SD = 0.17) (Appendix Ab). To compare t-DeepMGM sampled molecules with the collected known CB2 ligands, a t-SNE analysis based on MACCS fingerprints was conducted (Appendix A). The blue dots come from the initial half million ZINC compounds, which roughly define the overall chemical space. The black dots represent known CB2 ligands that concentrate at several regions within the boundary defined by the blue dots. The dots for the t-DeepMGM generated molecules are colored in green, red, purple, brown, and pink to reflect the different scaffolds used for sampling. The dots for the sampled molecules principally overlap with the central region of the orange dots. Starting with four scaffolds for sampling, generated molecules form a subset of known CB2 ligands. Likewise, with relatively limited structural diversity, CB2 target-specific ligands then contribute a subset to the overall chemical space. The transfer learning process forged the t-DeepMGM as to produce scaffold-focused chemical libraries with CB2 target preference. 

Again, four specific molecular descriptors, M.W., TPSA, SMR, and SlogP were selected for direct comparison between known CB2 ligands and t-DeepMGM generated molecules (Appendix A). In short, distributions of these four descriptors overlap well with mean values close to each other, indicative of the shared properties between the two groups of molecules. Generally, the distributions of descriptors for known CB2 ligands spread wider, with higher SD values as the generated molecules are less diverse with only four scaffolds.

### 3.5. Classification of Generated Molecules with a Discriminator

It is positive to notice that the t-DeepMGM generated molecules explore similar chemical space and share close physical–chemical properties to the known CB2 ligands, according to the t-SNE analysis and descriptor comparison. However, one more question remains at this stage: even though similar property profiles were shared, how can we tell whether generated molecules are active in an effective and efficient manner or not? To address this concern, a discriminator for the multilayer perceptron (MLP) was designed and constructed to be attached for assessment. Applying supervised machine learning classifiers is a well-accepted strategy in virtual screening, with successful stories reported [64,65,66,67]. The inclusion of the discriminator further closes the in silico design-test loop in this practice. 

Using the experimental CB2 *K_i_* value of 100 nM as the cutoff, collected CB2 ligands from ChEMBL were divided into 385 active and 564 inactive molecules. A total of 5000 randomly collected drug-like molecules from ZINC were mixed with inactive molecules to function as decoys. MACCS fingerprints were calculated for them. The discriminator was trained on fingerprints to distinguish active molecules from inactive ones and the decoys. The MLP discriminator was composed of three hidden layers with 166 hidden neurons for each layer. The six-fold cross-validation was performed for model generation and evaluation. The ROC curve (Appendix A), as well as a set of metrics including F1 score, accuracy, Cohen’s kappa, Matthews correlation coefficient (MCC), precision, and recall, were calculated to evaluate the model’s performance (Appendix A). 

The generated CB2 molecules from the t-DeepMGM from all five starting scaffolds were evaluated with the established MLP discriminator. The distribution of the predicted likelihood is summarized in Figure 5g. The majority of the t-DeepMGM generated CB2 molecules are predicted to have a high probability of being CB2 active. Interestingly, probability distributions of the generated molecules with different scaffolds can vary. Around 80% of generated molecules with the scaffolds of JTE-907, biamide inverse agonist, and triaryl sulfonamide derivatives have prediction scores between 0.9 to 1.0. In contrast, less than 40% of the generated molecules with the indole scaffold of the AM630 have a prediction score over 0.9. However, a percentage of 40 is still considered to be a high number for a constructed chemical library to be target-specific. 

### 3.6. Proof-of-Evidence in Discovering a Potential CB2 Negative Allosteric Modulator, XIE9137

With the development of the aforementioned in silico design-test circle by combining the DeepMGM with the target-specific Discriminator (Figure 6a), we took one more step to biologically measure the generated target-specific molecules through experimental bio-validations. We speculatively chose the sulfonyl scaffold as a newly designed chemical series, targeting CB2. After transfer learning using 12,904 allosteric modulators of Class A GPCRs (collected from ChEMBL), a t-DeepMGM was applied to explore the substitution groups to the alpha-carbon position of the carbonyl group for adding diversified amines (Figure 6b). By sampling 10,000 times, 476 unique molecules resulted (with a validity of 0.56 and a uniqueness of 0.09). The preliminary decision of selecting a hit molecule for chemistry synthesis and biological testing was based on: (1) whether that molecule is top-ranked by the defined discriminator, and (2) whether starting reagents are immediately available for chemistry synthesis. XIE9137 was one of the synthesized compounds (Figure 6c). The commercially available 1-methyl-1H-benzo[c][1,2]thiazin-4(3H)-one 2,2-dioxide was reacted with benzylamine and triethyl orthoformate in ethanol to give the target compound. 

To evaluate the functional activities of XIE9137 on the CB2 receptor, [^35^S]-GTPγS experiments were conducted with Epics Therapeutics’ membrane preparations overexpressing human CB2Rs (hCB2R). CP55940, a full agonist of CB2, was used as a positive control (Figure 6d). XIE9137 showed no activation or inhibition activity by itself (Figure 6e). To explore the possibility that the compound influences agonist-induced CB2 activity, its effect on CP55940 activation of CB2 [^35^S]-GTPγS binding was evaluated. A dose-response analysis for XIE9137 was performed in the absence or presence of CP55940 at EC_20_ concentration (to assess the enhancement of agonist activation) (Figure 6f) or CP55940 at EC_80_ concentration (to assess the inhibition of agonist activation) (Figure 6g). The CP55940-induced stimulation of [^35^S]-GTPγS CB2 binding was not enhanced by XIE9137 (Figure 6f). However, the inhibition of [^35^S]-GTPγS binding to CB2 was observed using an EC_80_ concentration of CP55940 (Figure 6g). The ability of CP55940 to stimulate [^35^S]-GTPγS binding to CB2Rs was partially inhibited by XIE9137 at 3 µM (28% of inhibition) and 10 µM (45% of inhibition). Since XIE9137 alone exerts no effects on CB2 activity, these results (summarized in Figure 6h) indicate that XIE9137 is either a neutral antagonist or a negative allosteric modulator (NAM).

With the aim to fully investigate the inhibitory effects of our compounds and to highlight different behaviors in other CB2 mediated signaling pathways, further biological studies are expected. It will be interesting to investigate (1) the CB2 binding of XIE9137, (2) the functional activities on the CB1 receptor, and (3) the effects on cAMP production, β-arrestin, and ion channels. With the aim to complete the analysis on a structure-activity relationship (SAR), a series of derivatives are expected to be synthesized to (1) enumerate amines connecting the alpha-carbon, (2) grow the tertiary amine, and (3) explore moieties to the phenyl ring. Further studies on biology, pharmacology, and medicinal chemistry will not be covered here, as this manuscript is more on the development of the DeepMGM method. The identification of a potential CB2 NAM in this section gives support to the effectiveness of applying DeepMGM on facilitating drug discovery.

## 4. Conclusions

The possibility of utilizing deep learning generative modeling to construct scaffold-focused chemical libraries was investigated in this study. A general molecule generation model (g-DeepMGM) was established after training and fine-tuning with a half million compounds collected from the ZINC database. Two sampling practices using indole and purine scaffolds for compounds generation were performed and discussed. To explore one step further, a target-specific molecule generation model (t-DeepMGM) was constructed after the transfer learning process of reported target-specific ligands. Using the CB2 receptor as an example, four known scaffolds were adapted for molecular sampling. The sampled outcomes share similar properties with known active compounds. Finally, a multilayer perceptron discriminator that distinguishes active CB2 ligands from inactive and random compounds was trained and attached to assess the results from the t-DeepMGM generation. The combination of the DeepMGM and the discriminator contributes to an in silico molecular design-test loop. The identification of a CB2 inhibitor (neutral antagonist or NAM) supports the effectiveness of applying DeepMGM on facilitating drug discovery. This case study is an example that shows the role that AI can play in this current stage in combination with experimental research. Following the discussion in Segler et al., it is even possible that the iteration of the molecule generation, scoring, and model retraining using the generated molecules with high scores can result in potential target specific molecules even without known active compounds to initiate the process. [18] Deep-learning generative modeling is a positive addition to well-established drug discovery approaches by bringing in new opportunities both on rational molecule design and virtual screening. Together with autonomous synthetic robots [68,69,70,71,72], it is fascinating to imagine a world that conducts early-stage drug design and synthesis in a fully automated manner. We are probably at the corner of an upcoming revolution of drug discovery in the AI era, and the good news is that we are witnessing the change. 

## Figures and Tables

**Figure 1 cells-11-00915-f001:**
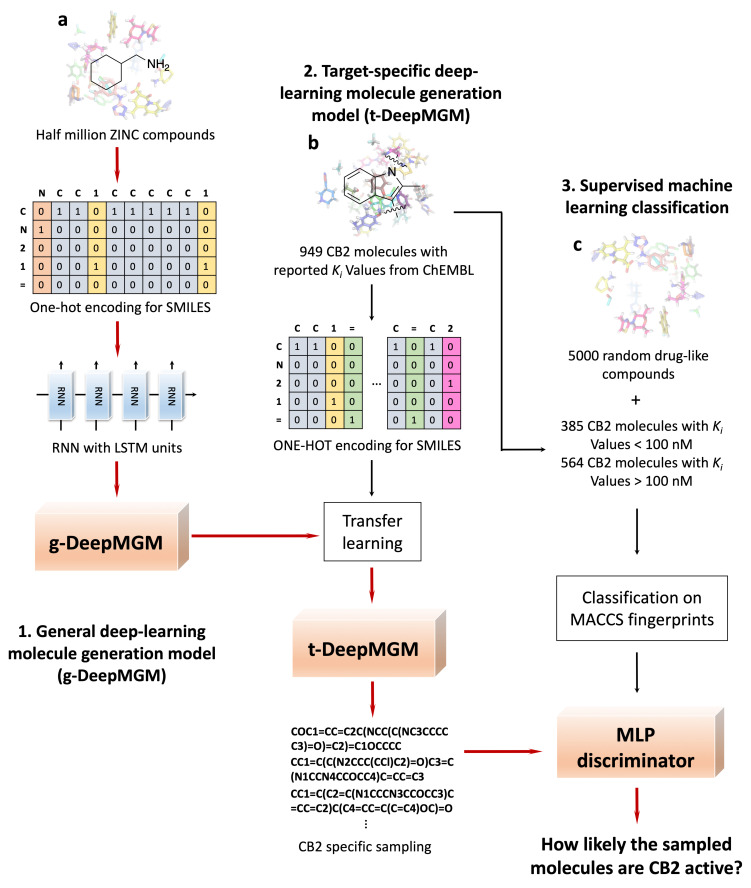
Schematic overview of the development pipeline for the general deep-learning molecule generation model, g-DeepMGM (**a**), and the target-specific deep-learning molecule generation model, t-DeepMGM (**b**) in combination with supervised machine-learning discriminator of multilayer perceptron (**c**).

**Figure 2 cells-11-00915-f002:**
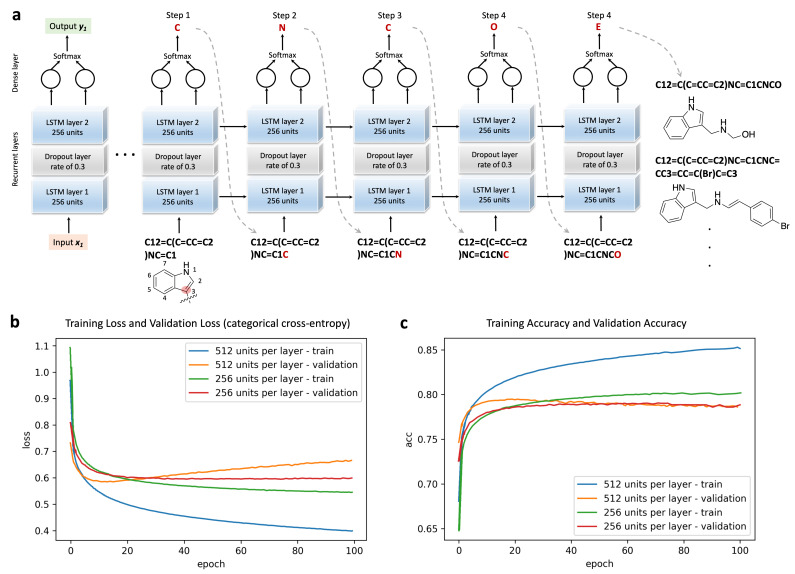
The training and sampling of the recurrent neural networks (RNN) model with long-short-term memory (LSTM) units. (**a**) The architecture of the neural network and the example sampling process with the indole ring as the input. (**b**) Training loss and validation loss at different training epochs. (**c**) Training accuracy and validation accuracy at different epochs.

**Figure 3 cells-11-00915-f003:**
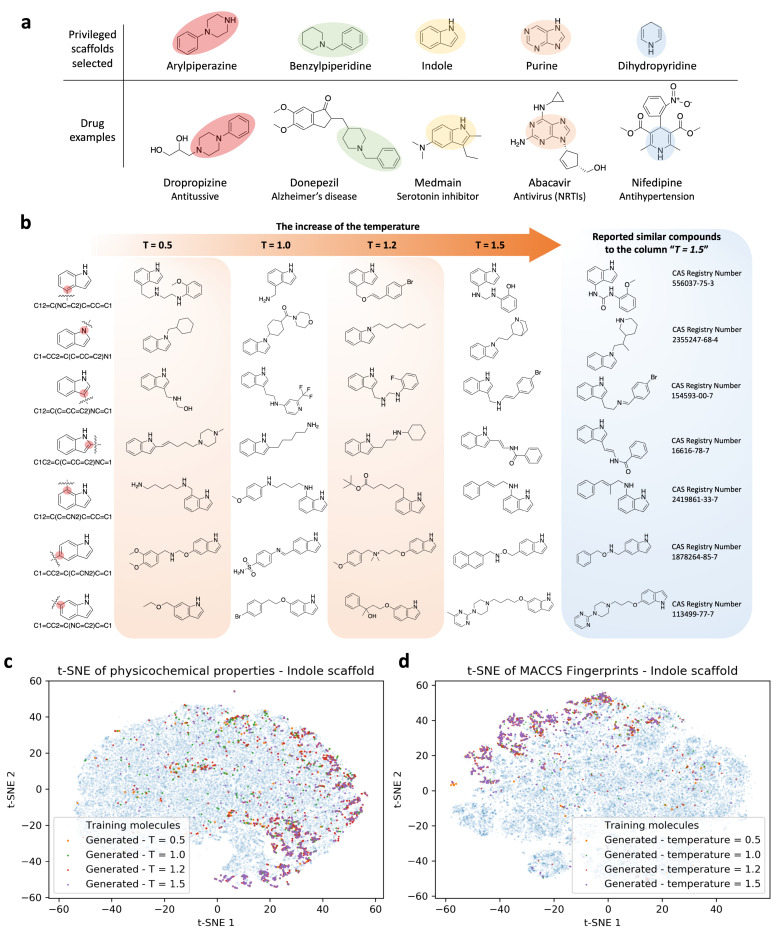
Privileged scaffolds and sampling examples of g-DeepMGM on indole. (**a**) Privileged scaffolds and the illustrated drugs that are derived from them. (**b**) Randomly selected sampling outcome for the indole scaffold under four temperatures with the g-DeepMGM at epoch 40. Seven SMILES strings representing seven addition positions on the indole were fed as the initial input. Reported similar compounds to the generated molecules in the column “T = 1.5” are listed for comparison. Using both physical-chemical properties-based (**c**) and MACCS fingerprints-based (**d**) t-SNE analysis to compare generated indole molecules and training compounds.

**Figure 4 cells-11-00915-f004:**
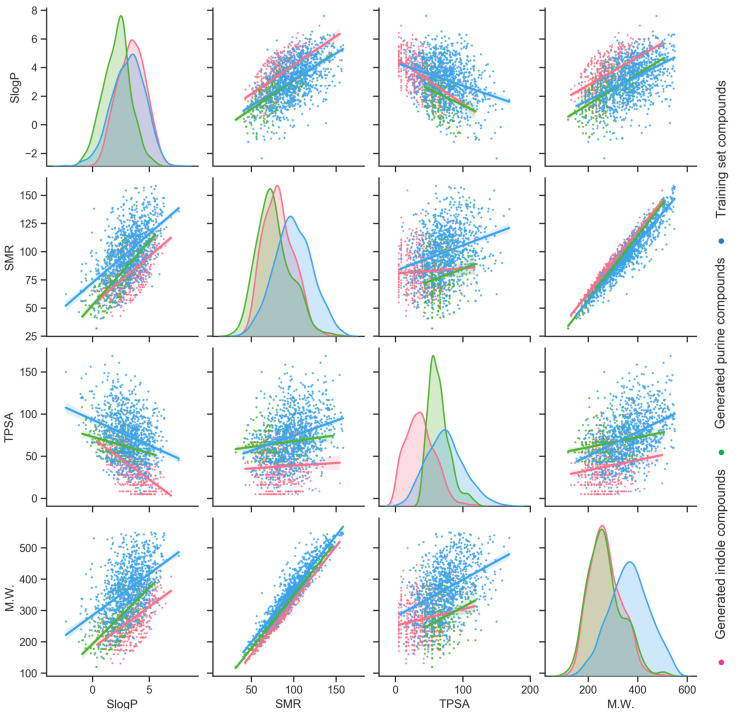
Distribution and correlation of molecular weight (M.W.), topological polar surface area (TPSA), molecular refractivity (SMR), and log of the octanol/water partition coefficient (SlogP) for training compounds, generated indole compounds, and generated purine compounds.

**Figure 5 cells-11-00915-f005:**
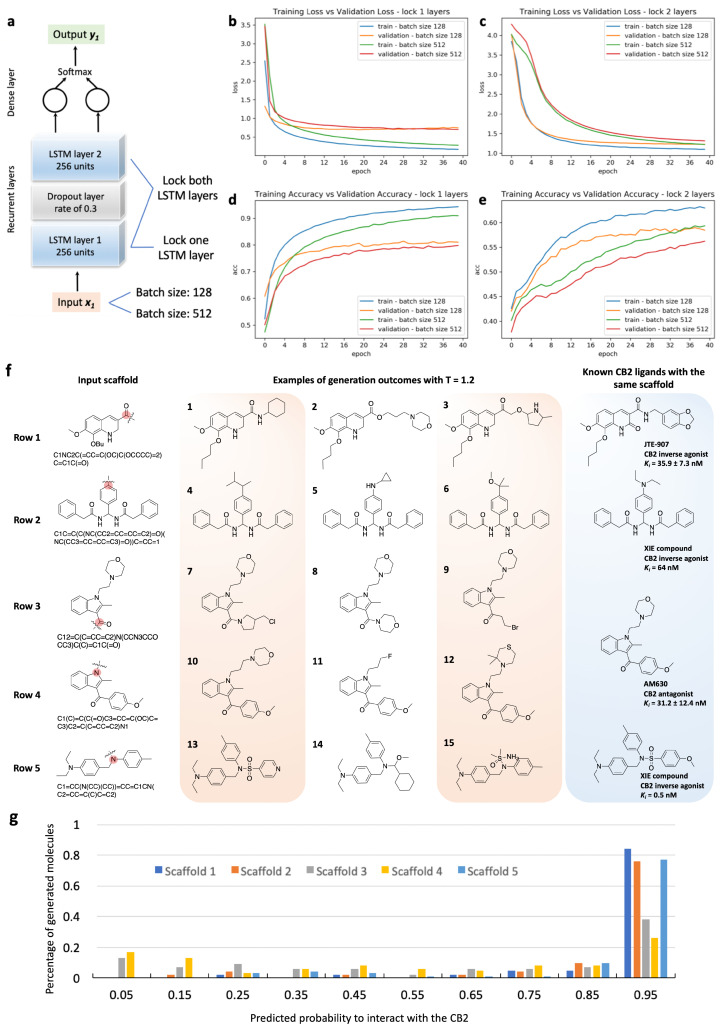
Developing t-DeepMGM to sample CB2 target-specific molecules after the transfer learning. (**a**) Architecture of the t-DeepMGM in transfer learning. The transfer learning processes with either one LSTM layer locked (**b**,**d**) or both LSTM layers locked (**c**,**e**) are performed for comparison. The influence of using different sizes of mini-batches (128 and 512) was also evaluated. (**f**) Molecular sampling examples from the t-DeepMGM for four known CB2 scaffolds. (**g**) The distribution of the predicted likelihood of generated CB2 molecules by the MLP discriminator.

**Figure 6 cells-11-00915-f006:**
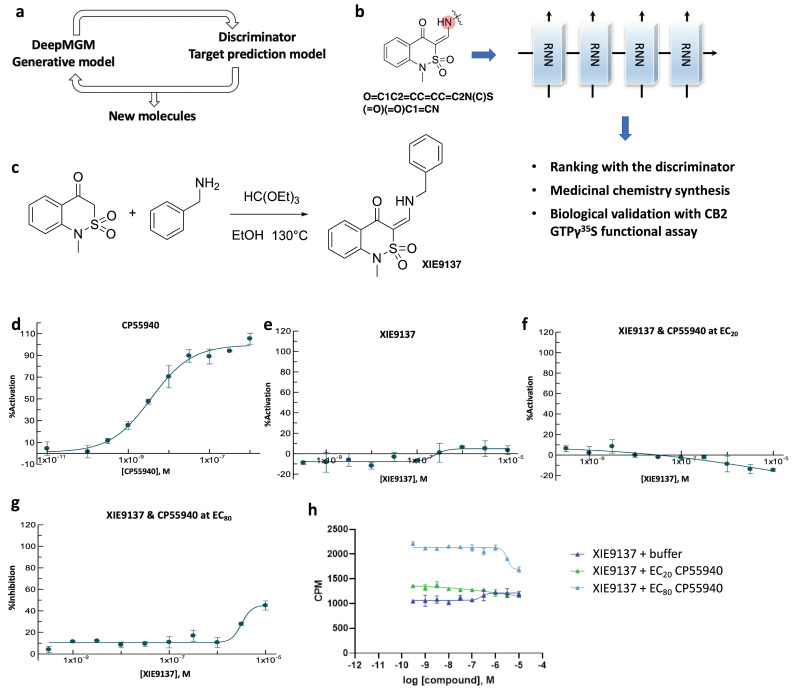
Identification of XIE9137 as a potential negative allosteric modulator for CB2. (**a**) Scheme of the in silico design-test circle. (**b**) Using chemistry synthesis and biological validation to evaluate generated molecules by the t-DeepMGM. (**c**) Synthetic route to obtain the target compound. (**d**,**e**) CB2 [^35^S]-GTPγS functional assay for CP55940 (**d**) and XIE9137 (**e**) alone. (**f**,**g**) CB2 [^35^S]-GTPγS functional assay for XIE9137 with the coexistence of CP55940 at EC_20_ concentration (**f**) and EC_80_ concentration (**g**). (**h**) Counts per minute in the CB2 [^35^S]-GTPγS functional assay for XIE9137 at different log concentrations.

**Table 1 cells-11-00915-t001:** The validity, uniqueness, and novelty of sampled indole molecules under different epochs and sampling temperatures of the g-DeepMGM.

Epoch	Temperature	Validity	Uniqueness	Novelty
**10** (training loss: 0.6309)	0.5	0.867	0.092	0.271
	**1.0**	**0.517**	**0.303**	**0.528**
	1.2	0.409	0.349	0.627
	1.5	0.313	0.357	0.719
**20** (training loss: 0.5966)	0.5	0.891	0.115	0.265
	**1.0**	**0.552**	**0.469**	**0.518**
	1.2	0.456	0.522	0.597
	1.5	0.342	0.508	0.706
**40** (training loss: 0.5700)	0.5	0.905	0.086	0.202
	**1.0**	**0.672**	**0.350**	**0.409**
	1.2	0.553	0.433	0.527
	1.5	0.399	0.476	0.653
**100** (training loss: 0.5444)	0.5	0.789	0.069	0.234
	**1.0**	**0.598**	**0.311**	**0.419**
	1.2	0.505	0.394	0.523
	1.5	0.381	0.455	0.629

## Data Availability

The 500,000 molecules were randomly collected from the ZINC database (https://zinc.docking.org), accessed on 14 November 2019. The 949 molecules with reported *K_i_* values for cannabinoid receptor 2 were collected from the ChEMBL database (https://www.ebi.ac.uk/chembl/), accessed on 21 January 2019. All chemical and biological resources are available upon request from the corresponding author.

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
