# Peer review of "Artificial Intelligent Deep Learning Molecular Generative Modeling of Scaffold-Focused and Cannabinoid CB2 Target-Specific Small-Molecule Sublibraries"

_cells, 2022, doi:10.3390/cells11050915_

Round 1
Reviewer 1 Report
Overall, the manuscript is well written and the rational was explained according to the experiments, which facilitate the comprehension. It is very original, innovative and used not only in silico experiments, but also experimental proof-of-concepts, which enrich the scientific merit.
The authors opted to use the ZINC database, which has a significant number of compounds (many works about artificial intelligence, with the same purpose, have used other databases). As the literature has some articles that use the same rational for drug discovery and new libraries obtention, authors should cite them in the introduction or discussion section, principally to compare the method presented here to others. In this sense, the introduction should be completed to bring more information regarding AI field for drug design. Moreover, considering a manuscript submitted to “Cells” and “Cell Signaling” section, information about the importance of the discovery of new CB2 inhibitors should be evidenced, justifying all the efforts to new molecules attainment.
The strategy to first select the molecules by the Lipinski’s rules is very interesting, because focus only on druggable compounds, which in the further step will make difference.
Moreover, the study was designed to have different compounds based not only on ADME proprieties, but also protein binding. In this sense, generated compounds had smaller molecular weight, SMR and TPSA values, compared to trained molecules. These features can represent advantages for protein binding and pharmacokinetics, depending on the target considered and its distribution through the organism.
Thus, the manuscript is suitable for publication, after including information in the introduction section.
Author Response
We sincerely thank reviewers for their constructive comments and suggestion. A point-by-point reply is attached.

Reviewer 2 Report
The authors presented a computational framework for bio-active molecule identification with deep learning techniques. Moreover, some experiments were conducted to validate the predictions. Generally speaking, this is a well-designed study. Thank you for the good work! Some minor changes are suggested to improve the manuscript quality.
- Author address information is not detailed listed. For example, Drug Discovery Institute and people would have no idea where it is and I would assume following the style of #4 is good.
- "de novo" should be italic throughout the text.
- In dataset preparation, 500, 000 molecules were randomly sampled from ZINC. But one would argue that having some prefilters would be a better choice, which includes PAINS filters, some drug-likeness filters.
- "Ki" in line 93 on page 2 should have "i" as a subscript.
- In Section 2.2, SMILES strings were used. But it is unclear that are the SMILES strings in canonical form or not, which is very important. Moreover, any stereochemistry information incorporated with isomeric SMILES?
- In line 165 on page 4, the abbreviation should be "QED" instead of "QES".
- References for software pages should be added, eg. RDKit, silicos_it.
- The source code for this model is not provided and it would be great to have that hosted in GitHub or other platforms.
Author Response

(The authors gave the same response as above.)
